# ER Unfolded Protein Response in Liver In Vivo Is Characterized by Reduced, Not Increased, De Novo Lipogenesis and Cholesterol Synthesis Rates with Uptake of Fatty Acids from Adipose Tissue: Integrated Gene Expression, Translation Rates and Metabolic Fluxes

**DOI:** 10.3390/ijms23031073

**Published:** 2022-01-19

**Authors:** Catherine P. Ward, Lucy Peng, Samuel Yuen, Michael Chang, Rozalina Karapetyan, Edna Nyangau, Hussein Mohammed, Hector Palacios, Naveed Ziari, Larry K. Joe, Ashley E. Frakes, Mohamad Dandan, Andrew Dillin, Marc K. Hellerstein

**Affiliations:** 1Department of Nutritional Sciences and Toxicology, University of California, Berkeley, CA 92093, USA; cschneider@berkeley.edu (C.P.W.); lucyzpeng@berkeley.edu (L.P.); samyuen2@berkeley.edu (S.Y.); mikebklklk@berkeley.edu (M.C.); rozalina@berkeley.edu (R.K.); edna.n@berkeley.edu (E.N.); mhussein@berkeley.edu (H.M.); palacioshh@berkeley.edu (H.P.); naveedziari@berkeley.edu (N.Z.); mohamad_dandan@berkeley.edu (M.D.); 2Department of Molecular and Cellular Biology, University of California, Berkeley, CA 92093, USA; larry.joe@berkeley.edu (L.K.J.); afrakes@berkeley.edu (A.E.F.); dillin@berkeley.edu (A.D.)

**Keywords:** ER stress, proteostasis, UPR, lipid metabolism, metabolism

## Abstract

The unfolded protein response in the endoplasmic reticulum (UPR^ER^) is involved in a number of metabolic diseases. Here, we characterize UPR^ER^-induced metabolic changes in mouse livers in vivo through metabolic labeling and mass spectrometric analysis of lipid and proteome-wide fluxes. We induced UPR^ER^ by tunicamycin administration and measured synthesis rates of proteins, fatty acids and cholesterol, as well as RNA-seq. Contrary to reports in isolated cells, hepatic de novo lipogenesis and cholesterogenesis were markedly reduced, as were mRNA levels and synthesis rates of lipogenic proteins. H&E staining showed enrichment with lipid droplets while electron microscopy revealed ER morphological changes. Interestingly, the pre-labeling of adipose tissue prior to UPR^ER^ induction resulted in the redistribution of labeled fatty acids from adipose tissue to the liver, with replacement by unlabeled glycerol in the liver acylglycerides, indicating that the liver uptake was of free fatty acids, not whole glycerolipids. The redistribution of adipose fatty acids to the liver was not explicable by altered plasma insulin, increased fatty acid levels (lipolysis) or by reduced food intake. Synthesis of most liver proteins was suppressed under UPR^ER^ conditions, with the exception of BiP, other chaperones, protein disulfide isomerases, and proteins of ribosomal biogenesis. Protein synthesis rates generally, but not always, paralleled changes in mRNA. In summary, this combined approach, linking static changes with fluxes, revealed an integrated reduction of lipid and cholesterol synthesis pathways, from gene expression to translation and metabolic flux rates, under UPR^ER^ conditions. The reduced lipogenesis does not parallel human fatty liver disease. This approach provides powerful tools to characterize metabolic processes underlying hepatic UPR^ER^ in vivo.

## 1. Background

Proteostasis, or protein homeostasis, is important in maintaining a healthy cellular environment under stressful conditions [1]. The accumulation of misfolded proteins can perturb proteostasis, and thereby initiate the endoplasmic reticulum unfolded protein response (UPR^ER^) [2,3,4]. The role of the UPR^ER^ in metabolic diseases has been hypothesized to be due to dysregulation of lipid homeostasis, but remains poorly understood [5,6,7,8]. Hepatic steatosis, one metabolic disease questioned for UPR^ER^ involvement, is characterized by lipid accumulation in the liver, eventually leading to liver cirrhosis if unresolved [6]. Here, our aim was to characterize metabolic flux changes of UPR^ER^, including metabolic sources of deposited lipids in the liver, as well as changes in protein synthesis rates across the proteome in relation to changes in gene expression (mRNA levels).

The UPR^ER^ consists of three arms, led by ER membrane anchored inositol requiring enzyme 1α (IRE1ɑ), PKR-like ER kinase (PERK), and activating transcription factor 6 (ATF6) [9]. In times of UPR^ER^, a binding-immunoglobulin protein, or GRP-78, (BiP), a key chaperone, moves away from the ER membrane to combat the accumulation of misfolded proteins and activates these three arms [9]. A hallmark of the UPR^ER^ is suppressed global protein translation, with the exception of key ER stress responders, such as chaperones [10]. Although these responses have been characterized at the level of mRNA and protein concentrations, rates of protein synthesis (translation rates) in response to the induction of the UPR^ER^ in vivo have yet to be characterized. Nor has the relation between mRNA changes and protein synthesis at different points after UPR^ER^ induction been established. Dynamic proteomic techniques that we have developed [11], using heavy water labeling and tandem mass spectrometric kinetic analysis based on combinatorial analysis of deuterium incorporation patterns, can be applied to measure hepatic protein translation rates after tunicamycin-induced UPR^ER^ in living mice.

Another widely believed hallmark of the UPR^ER^ is an increase of de novo lipogenesis to explain the expansion of ER lipids [12,13]. This has, however, been studied mostly in cell culture models [14]. New fatty acid synthesis in the liver is inconsistent, however, with the reduced lipogenic gene expression in vivo in the livers of mice under UPR^ER^ [15,16]. Lipid accumulation in the liver is seen in mice under conditions of ER stress, yet the source of these lipids was unknown^6^. Increased hepatic lipids under UPR^ER^ conditions may be important in providing an added surface for the resolution of ER protein synthetic stress [17], but other metabolic sources in a whole organism that are not available or apparent in isolated cell systems could be responsible for increases in ER lipids, including imports from other tissues.

We were curious about how rates of de novo lipogenesis change after ER stress induction in vivo and how these might correlate with changes in protein translation rates. Accordingly, we used heavy water labeling to measure rates of de novo lipogenesis, cholesterol synthesis, and protein synthesis rates across the proteome in the liver over a three-day labeling period after tunicamycin-induced UPR^ER^. We also performed RNA-seq on the liver tissue to compare mRNA levels to rates of protein synthesis and lipogenic pathway fluxes. The purpose of our studies was to elucidate changes in proteome-wide changes in translation and lipid metabolism under UPR^ER^ activation. During our studies, we sought to determine the source of accumulated hepatic lipids to better understand the potential involvement of the UPR^ER^ in metabolic diseases.

## 2. Results

### 2.1. RNA-Seq of Mouse Liver under Induced UPR^ER^ Reveals Decreased Expression of Genes Involved in Lipid and Cholesterol Synthesis as Well as Many Downregulated and Some Upregulated Gene Ontologies

To investigate the time period following the initiation of the UPR^ER^, we used tunicamycin to induce UPR^ER^ in mice and took liver samples over the subsequent three days (Figure 1a). Tunicamycin, a common drug used to activate the UPR^ER^, inhibits N-linked glycosylation, leading to the accumulation of misfolded proteins, and was chosen due to its ability to initiate all three arms of the UPR^ER^. RNA-seq of the liver tissue revealed many significant changes, including the decreased expression of genes involved in lipid synthesis and cholesterol metabolism. Upregulated ontologies included genes involved in the response to ER stress, ER associated degradation (ERAD), and ribosome biogenesis (Figure 1b–g). The expression of genes involved in the UPR^ER^ shifted over time post ER-stress induction (Figure 1b–f).

### 2.2. Dynamic Proteomic Measurements Reveal Decreased Global Protein Synthesis Rates, including Those Involved in Lipid Synthesis, but Not for Key UPR^ER^ Proteins

We asked whether protein translation rates match the canonical changes in mRNA levels, by using the dynamic proteomic approach [11] that we have previously described. Deuterated water was administered concurrently with tunicamycin to label proteins that were translated over the three-day treatment period after UPR^ER^ induction (Appendix A). During the first 12 h post-tunicamycin, synthesis rates were suppressed for most proteins measured, with the exception of proteins involved in UPR^ER^, including BiP, protein disulfide isomerases, and other chaperones (Figure 2a–f). Proteins for which translation rates were significantly increased or decreased with the induction of UPR^ER^, as compared to controls, were organized into KEGG-pathways (Kyoto Encyclopedia of Genes and Genomes) for analysis (Figure 2g). The synthesis of proteins involved in fatty acid metabolism were decreased, starting at 6 h post tunicamycin treatment (Figure 2g), as were most proteins characterized as being involved in “metabolic pathways”. By 12 h post treatment, some protein translation rates began to recover, while others remained reduced through to 48 h. Most protein synthesis rates returned to baseline values at the final time point measured, 72 h (Figure 2g). Correlations between mRNA levels and protein synthesis rates revealed many correlations, but divergences as well (e.g., lack of return to baseline values by 72 h).

KEGG-pathway analysis revealed significantly increased synthesis rates of proteins involved in protein processing in the ER from the 12 h time point continuing to the 72 h time point, which can be attributed to the many chaperones and other UPR^ER^ responders. In particular, BiP showed a marked increase in synthesis rate at earlier time points, and though trending down over time, remained elevated at 72 h (Figure 2c). KEGG pathway analysis showed markedly decreased synthesis rates of proteins involved in lipid metabolism at 24 and 48 h (Figure 2g). The decreased synthesis of proteins involved in glutathione synthesis matched the decline observed in the RNA-seq data (Figure 1 and Figure 2g). Proteins involved in ribosomal biogenesis were also increased at the 48 h timepoint, matching the RNA-seq data (Figure 1 and Figure 2g).

### 2.3. UPR^ER^ Induction Causes Lipid Accumulation by Histology

Hematoxylin and eosin (H&E) staining of livers, taken from mice treated with tunicamycin or DMSO, revealed lipid accumulation in the liver starting at 48 h post UPR^ER^ induction. Lipids are indicated by a lack of stain, thus appear as white in the images. Earlier time points revealed no lipid differences compared to controls (Figure 3b).

### 2.4. Changes in ER Morphology by Electron Microscopy

We asked how the decline of gene expression and the protein synthesis rates of lipid synthesis proteins correlated with ER membrane expansion, reported by other groups, under ER stress conditions [17]. Electron microscopy revealed distinct morphological changes to ER structure, beginning at 12 h post-tunicamycin treatment and continuing through to the 72 h endpoint. ER in the control samples presented as stacked and ribosome studded, as expected, whereas the ER observed in the tunicamycin-treated animals were disorganized and appeared like bubbles, or cobblestones, and appeared to be barren of the usual ribosomes (Figure 3a).

### 2.5. Lipid and Cholesterol Synthesis Rates Are Decreased Post-UPR^ER^ Induction

Due to the striking morphological changes, we asked if lipids in the newly expanded ER membrane and droplets were coming from de novo fatty acid synthesis in the liver, as has been reported in cultured liver cells [13]. This was measured by the heavy water labeling of fatty acids in phospholipids and triglycerides, and in free and esterified cholesterol (Appendix A) [18]. The contribution from de novo lipogenesis to palmitate in both hepatic triglycerides and phospholipids was significantly decreased in the beginning of 48 h post tunicamycin treatment, and continued to be reduced through the last time point at 72 h (Figure 4a,b), consistent with the reduced mRNA levels and translation rates for lipogenic proteins. De novo synthesis of both free and esterified cholesterol in the liver was also significantly decreased at 48 and 72 h post ER stress initiation (Figure 4c), consistent with the significant decline in expression of cholesterol synthesis genes and lower translation rates of cholesterogenic proteins in the liver.

### 2.6. Mobilization of Fatty Acids from Adipose Tissue to Liver Is Increased under UPR^ER^ Conditions

To answer the question of where lipids in the liver were coming from, if not from de novo synthesis, we pre-labeled extrahepatic triglycerides (i.e., primarily adipose tissue lipid stores) prior to inducing UPR^ER^. Mice were given deuterated water for 7 weeks to label newly synthesized fatty acids in the adipose tissue. We then discontinued heavy water intake for 2 weeks, which is sufficient time to allow deuterium enrichment to die away in body water and in liver triglycerides, which have much shorter half-lives than triglycerides in adipose tissue [17] (Figure 5a,b and Appendix A). After tunicamycin-induced UPR^ER^, deuterium enrichment of the palmitate in hepatic triglycerides increased, whereas deuterium enrichment of the palmitate in adipose tissue triglycerides decreased compared to controls (Figure 5c), representing lipid mobilization from adipose tissue to the liver. A decline in triglyceride-glycerol enrichment, concurrently with increased palmitate enrichment in the liver and a reduction in enrichment of glycerol in phospholipids in the liver, indicate that intact triglycerides and phospholipids had turned over to the free glycerol level under UPR^ER^ conditions (Figure 5d). This can only be explained by the uptake of palmitate in the form of free fatty acids, as opposed to the transport of whole triglycerides or phospholipids from the uptake of lipoproteins, including lipids from dietary-derived lipoproteins which would dilute both the fatty acid and the glycerol moieties of acyglycerides in the liver (Figure 5c,d).

### 2.7. Pair-Feeding and Measurement of Plasma Insulin and Free Fatty Acid Concentrations

To evaluate the potential role of tunicamycin-induced anorexia, the altered plasma levels of insulin or free fatty acids in the redistribution of fatty acids from adipose tissue to liver, we performed pair-feeding studies and measured plasma insulin and free fatty acid concentrations. Mice were sacrificed in a fed state, which represents the condition in which most lipid storage in the liver is likely to occur, whereas lipid oxidation predominates under fasting conditions. If increased serum insulin levels or free fatty acid levels drive lipid storage, this should be reflected in the fed state. We observed no significant differences, however, between control, tunicamycin-treated, and pair-fed (to tunicamycin) groups for plasma insulin or free fatty acid concentrations (Figure 6a–c). Plasma non-esterified fatty acid concentrations are generally well correlated with the rates of adipose tissue lipolysis and the release of fatty acids [19,20], so these findings exclude both the altered lipolytic rates induced by lower insulin levels, or alterations in the insulin-induced uptake and esterification of plasma fatty acids by the liver as explanations for redistribution of fatty acids from adipose tissue to the liver during UPR^ER^.

## 3. Discussion

We report here a striking decline in de novo lipogenesis by all these metrics: reduced flux rates through de novo lipogenesis and cholesterol synthesis pathways, reduced synthesis rates of lipogenic and cholesterogenic proteins, and reduced mRNA levels for lipid-synthesis related genes. However, electron microscopy, and hematoxylin and eosin staining visualized lipid accumulation and changes in the ER membrane morphology, over this time. To explain the metabolic source of hepatic lipid accumulation, we pre-labeled adipose triglycerides by long-term heavy water administration, and allowed deuterium enrichment to die away in body water and in liver triglycerides prior to UPR^ER^ induction. We demonstrated for the first time that the lipids that accumulate in the liver during UPR^ER^ in vivo are mobilized from other tissues (adipose fatty acids), not derived from hepatic de novo lipogenesis, and that the redistribution of fatty acids to the liver is not explicable by altered plasma insulin levels, adipose lipolysis or food intake.

Metabolic responses to the initiation of the UPR^ER^ are not well understood [21]. We used metabolic labeling with stable isotopes to concurrently measure synthesis rates of fatty acids, cholesterol, and proteins across multiple ontologies after inducing UPR^ER^, with the goal of characterizing metabolic flux signatures over time at multiple levels of control, including gene expression patterns, and their relation to ultrastructural changes.

Our most striking finding was that, contrary to reported findings in isolated cells [17,22,23,24], rates of hepatic de novo lipogenesis and cholesterogenesis were markedly reduced, not increased, during UPR^ER^. Lower de novo lipogenesis and cholesterol synthesis rates are consistent with the reduced synthesis rates and mRNA levels of lipogenic and cholesterogenic proteins that we also measured. These reductions in de novo lipid and cholesterol synthesis rates, gene expression, and protein synthesis rates were remarkable in view of the well-established changes in the ER membrane structure and lipid stores [11,17,22,23,24] during the UPR^ER^. Our ultrastructural observations confirmed that after 12 h of UPR^ER^ induction, ER membranes appeared strikingly different, by electron microscopy, and bubble-like, consistent with expansion [17,22,23,24], supported by H&E staining that showed enrichment with lipid droplets.

To understand the source of added liver lipids, we carried out long-term pre-labeling of lipids in vivo for 7 weeks, followed by 2 weeks of wash-out prior to UPR^ER^ induction, to allow deuterium enrichment to die away in body water and liver triglycerides. This approach is possible because liver triglycerides have much shorter half-lives than triglycerides in adipose tissue [19]. We observed a significantly greater mobilization of labeled lipids from the adipose tissue to the liver during UPR^ER^, but also the replacement of the glycerol moiety by unlabeled glycerol in liver lipids (Figure 5a,b). The observation of concurrent increases in labeled palmitate from adipose, but with the glycerol moiety unlabeled, indicated that the hepatic uptake was of adipose-derived free fatty acids, not whole glycerolipids (triglycerides or phospholipids). Diet-derived glycerol-lipids taken up as chylomicron remnants, therefore, cannot explain the accrual of liver lipids in vivo during UPR^ER^, as this would have resulted in unlabeled glycerol and fatty acid moieties in acylglycerides.

Moreover, comparisons of the control, tunicamycin-treated and pair-fed animals revealed no differences in plasma insulin or fatty acid levels. Tunicamycin-induced anorexia, as previously characterized by other groups [15], was observed in our studies. Mice treated with tunicamycin ate and weighed less than control mice, and presented with less adipose tissue upon dissection, supporting the observed mobilization of lipids from adipose to the liver. In contrast, reduced food intake alone (pair-feeding) does not induce alterations or expansion of liver ER membranes or lipid accumulation in the liver [25,26].

These results in living mice differ from some studies in isolated cells [14,27,28], which have reported an increase in the expression of genes involved in de novo lipid synthesis during the UPR^ER^. It is also worth noting that non-alcoholic fatty liver disease in humans is characterized by markedly elevated rates of de novo hepatic lipogenesis [29,30,31] and cholesterol synthesis, contrary to our findings in the mouse liver during UPR^ER^. Although our studies are in mice, these findings suggest that UPR^ER^ may not be a primary contributor to lipid accrual in human non-alcoholic fatty liver disease. It has been often noted that there is a publication bias toward positive results [32] which can result in misdirected efforts. The finding here, that UPR^ER^ in mouse livers in vivo does not reflect the pattern of fatty acid fluxes in non-alcoholic fatty liver disease in humans in vivo, is a potentially important result. Direct confirmation by the measurement of de novo lipogenesis and cholesterol synthesis in humans in states of hepatic UPR^ER^ will be of interest.

We were able to exclude some obvious potential metabolic mechanisms underlying the redistribution of adipose tissue fatty acids to the liver during UPR^ER^ in the whole animal. Reduced plasma insulin levels, causing higher rates of adipose tissue lipolysis, in turn, induced by reduced food intake, was excluded as a possibility by the pair-feeding data and the absence of changes in plasma insulin or free fatty acid concentrations during tunicamycin–induced UPR^ER^. An alternative explanation, that increases in insulin levels induce higher esterification rates of plasma free fatty acids by the liver, is also not tenable. The possible basis for increased avidity and/or the deposition efficiency of circulating free fatty acids into liver glycerolipids remains unknown, and will be of interest to pursue in future studies of hepatic UPR^ER^.

Overall, proteome-wide protein synthesis rates declined with tunicamycin-induced ER stress, with the exception of chaperones and other key ER proteins recognized to be induced during the UPR^ER^ [33]. Protein synthesis rates generally matched the signatures measured through RNA-seq, which were similar to canonical UPR^ER^ signatures reported previously [34,35], but there were exceptions to this relationship. These exceptions include the continued elevation of UPR^ER^ involved genes at 72 h, whereas the actual protein translation rates had trended back towards a normal signature. This discrepancy may be explained by lags in mRNA turnover, whereas protein translation rates reflect the real-time flux for each protein. Correlations between mRNA expression and protein levels or synthesis rates (i.e., between transcription and translation) may be modified by translational control [36] or other factors, and should not be taken for granted [36,37,38]. We have reported in skeletal muscle, for example, clear dissociations between mRNA levels and protein synthesis rates [39]. Our data here are the first to show that global proteome-wide protein synthesis rates across multiple gene ontologies are reduced in vivo in the liver during UPR^ER^ and generally, but not always, correlate with mRNA changes in this setting.

More specifically, we found that synthesis rates of hepatic proteins involved in lipid metabolism and cholesterol synthesis were decreased in response to tunicamycin. In combination with similar reductions in gene expression for lipogenic proteins and flux rates through these lipogenic pathways, our results reveal a consistent correlation of transcription, translation and metabolic flux rates for the pathways of de novo synthesis of hepatic lipids (palmitate in triglycerides and phospholipids and free and esterified cholesterol) during UPR^ER^ in vivo.

Interestingly, kidney tissue from the same tunicamycin treated mice failed to present a similar protein synthesis signature, indicating tissue specific activation of UPR^ER^ cascades after whole organism tunicamycin treatment (Appendix A).

In summary, these data add to our knowledge about the metabolic alterations present during the UPR^ER^. Lipids taken up from outside the liver (derived from adipose tissue fatty acids), not lipids synthesized de novo locally, are the source of accumulated lipids in the liver during UPR^ER^ in vivo. Key metabolic pathways of lipid and cholesterol synthesis are reduced in an integrated and consistent manner, from gene expression to translation rates and metabolic pathway fluxes, while other pathways are perturbed in complex and not entirely predictable ways, including the mobilization of lipids from adipose tissue to the liver. These findings provide a differential metabolic signature of UPR^ER^ and contrast to hepatic lipid accumulation pathways in human fatty liver disease, for example, where de novo lipogenesis and cholesterogenesis are greatly elevated [29,30,31].

## 4. Methods

### 4.1. Animals

C57BL/6J mice acquired from The Jackson Laboratory were used for this study. Mice were male and 12 weeks of age. All mice were housed according to the Animal Care and Use Committee (ACUC) standards in the animal facility at UC Berkeley. Mice were fed a standard chow diet and water ad libitum.

### 4.2. Deuterated Water Labeling and Tunicamycin Treatment in Mice

Mice were labeled with deuterated water (heavy water, ^2^H_2_O), beginning at time point 0 (t^0^), through to the end of the experiment. Five mice were used per group. Proteins synthesized after t^0^ incorporated deuterium-labeled amino acids, thus enabling the measurement of proteins synthesized during the period of exposure to heavy water [11,40]. Deuterium is rapidly incorporated throughout the body of an organism after treatment, bringing the deuterium enrichment in body water up to 5%. Deuterium enrichment is maintained through the intake of 8% ^2^H_2_O given as drinking water, thus making it an optimal labeling approach for in vivo experimental study. Mice are injected intraperitoneally (IP) with 100% ^2^H_2_O, containing either tunicamycin dissolved in DMSO, or DMSO control. Mice were treated with 1.5 mg/kg tunicamycin dissolved in DMSO at t^0^, or DMSO control, and tissues were harvested 6, 12, 24, 48, and 72 h after the initial injection (*n* = 5 mice per group).

### 4.3. Deuterated Water Labeling and Tunicamycin Treatment in Mice: Pre-Label of Adipose Tissue Triglycerides

Mice were labeled with deuterated water (heavy water, ^2^H_2_O) for 7 weeks to saturate tissues with deuterium in vivo. Deuterium was rapidly incorporated throughout the body of an organism after treatment, bringing the deuterium enrichment in body water up to 5%. Deuterium enrichment was maintained through the intake of 8% ^2^H_2_O, given as drinking water, thus making it an optimal labeling approach for a long-term in vivo experimental study. Mice were then given non-labeling drinking water to wash deuterium label out of faster generating tissues (i.e., the liver), but not enough time to significantly reduce label in slower lipid turnover tissues such as the adipose. After 2 weeks, mice were injected intraperitoneally (IP) with either tunicamycin dissolved in DMSO (at 10 mg/mL), or DMSO control. Mice were treated with 1.5 mg/kg of tunicamycin dissolved in DMSO, or DMSO control of equal volume, and tissues were harvested 72 h after the initial injection (*n* = 4 mice per group).

### 4.4. Body Water Enrichment Analysis

Mouse livers were distilled overnight upside down on a bead bath at 85 °C to evaporate out body water. Deuterium present in the body water were exchanged onto acetone, and deuterium enrichment in the body water was measured via gas chromatography mass spectrometry (GC-MS) [19,41].

### 4.5. Tissue Preparation for Liquid Chromatography-Mass Spectrometry (LC-MS)

Tissues were flash frozen after harvest and homogenized in homogenization buffer (100 mM PMSF, 500 mM EDTA, EDTA-free Protease Inhibitor Cocktail (Roche, Basel, Switzerland, catalog number 11836170001), PBS) using a 5 mm stainless steel bead at 30 hertz for 45 s in a TissueLyser II (Qiagen, Hilden, Germany). Samples were then centrifuged at 10,000 rcf for 10 min at 4 °C. The supernatant was saved and protein was quantified using a Pierce BCA protein assay kit (ThermoFisher, catalog number 23225). One hundred micrograms of protein were used per sample. Twenty-five microliters of 100 mM ammonium bicarbonate solution, 25 µL TFE, and 2.3 µL of 200 mM DTT were added to each sample and incubated at 60 °C for 1 h. Ten microliters of 200 mM iodoacetamide were then added to each sample and allowed to incubate at room temperature in the dark for 1 h. Two microliters of 200 mM DTT was added and samples were incubated for 20 min in the dark. Each sample was then diluted with 300 µL of H_2_O and 100 µL of 100 mM ammonium bicarbonate solution. Trypsin was added at a ratio of 1:50 trypsin to protein (trypsin from porcine pancreas, Sigma Aldrich, catalog number T6567). Samples were incubated at 37 °C overnight. The next day, 2 µL of formic acid was added. Samples were centrifuged at 10,000 rcf for 10 min, collecting the supernatant. Supernatant was dried in a speedvac and re-suspended in 50 µL of 0.1% formic acid/3% acetonitrile/96.9% LC-MS grade water and transferred to LC-MS vials to be analyzed via LC-MS.

### 4.6. Liquid Chromatography-Mass Spectrometry (LC-MS) Analysis

Trypsin-digested peptides were analyzed on a 6550 quadrupole time of flight (Q-ToF) mass spectrometer, equipped with Chip Cube nano ESI source (Agilent Technologies, Santa Clara, CA, USA). High performance liquid chromatography (HPLC) separated the peptides using capillary and nano binary flow. Mobile phases were 95% acetonitrile/0.1% formic acid in LC-MS grade water. Peptides were eluted at 350 nL/minute flow rate with an 18 min LC gradient. Each sample was analyzed once for protein/peptide identification in data-dependent MS/MS mode and once for peptide isotope analysis in MS mode. Acquired MS/MS spectra were extracted and searched using Spectrum Mill Proteomics Workbench software (Agilent Technologies) and a mouse protein database (www.uniprot.org). Search results were validated with a global false discovery rate of 1%. A filtered list of peptides was collapsed into a nonredundant peptide formula database containing peptide elemental composition, mass, and retention time. This was used to extract mass isotope abundances (M0–M3) of each peptide from MS-only acquisition files with Mass Hunter Qualitative Analysis software (Agilent Technologies). Mass isotopomer distribution analysis (MIDA) [11,42,43] was used to calculate peptide elemental composition and curve-fit parameters for predicting peptide isotope enrichment based on precursor body water enrichment (p) and the number (n) of amino acids at C-H positions per peptide actively incorporating hydrogen (H) and deuterium (D) from body water. Subsequent data handling was performed using python-based scripts, with input of precursor body water enrichment for each subject, to yield fractional synthesis rate (FSR) data at the protein level. FSR data were filtered to exclude protein measurements with fewer than 2 peptide isotope measurements per protein. Details of FSR calculations and data filtering criteria have been described in detail previously [11,40].

### 4.7. Calculation of Fractional Replacement (f) and Replacement Rate Constant (k) for Individual Proteins

Details of f calculations were previously described [11,40]. These values were used to generate the ratio of tunicamycin treated to untreated synthesis rates.

### 4.8. Statistical Analysis

Data were analyzed using GraphPad Prism software (version 8.0). We used two-tailed *t*-tests and ANOVA tests to assess statistical analysis across groups.

### 4.9. KEGG Pathway Analysis

Protein fractional synthesis rates were weighted by the peptide count and averaged according to their KEGG pathway involvements. We used the Uniprot.ws package in R from Bioconductor to find mappings between UniProt accession numbers and their corresponding KEGG IDs for each protein. Tables were generated for the entire known proteome for mouse. We then used the Bio.KEGG module of Biopython in Python to access to the REST API of the KEGG database to get a list of pathways to which each protein belongs. A set of all the pathways relevant to the experiment was generated and each protein and its corresponding fold change value were assigned to each pathway. KEGG pathways with no less than five proteins were used for representation of the data.

### 4.10. Tissue Preparation for Gas Chromatography-Mass Spectrometry (GC-MS)

A chloroform methanol extraction was used to isolate lipids from the liver tissue. These lipids were run on a thin-layer chromatography (TLC) plate to separate phospholipid and triglyceride fractions. These fractions containing the palmitate were further derivatized for GC-MS analysis.

### 4.11. Gas Chromatography-Mass Spectrometry (GC-MS) Analysis

Palmitate and cholesterol isotopic enrichments were measured by GC-MS (Agilent models 6890 and 5973; Agilent, Inc., Santa Clara, CA, USA) using helium carrier gas, a DB-225 (DB 17 for cholesterol and DB 225 for palmitates) fused silica column (30 M × 0.25 mm ID × 0.25 µm), electron ionization mode, and monitoring *m/z* 385, 386, and 387 for palmitates, and 368, 369, and 370 for cholesterol acetyl derivatives, for M0, M1, and M2 respectively, as previously described [44,45]. Palmitate methyl ester enrichments were determined by GC-MS using a DB-17 column (30 M × 0.25 mm ID × 0.25 um), with helium as carrier gas, electron ionization mode, and monitoring *m/z* 270, 271, and 272 for M0, M1, and M2. Baseline unenriched standards for both analytes were measured concurrently to correct for abundance sensitivity.

### 4.12. Calculation of De Novo Lipogenesis (DNL) and Cholesterol Synthesis

The measurement of newly synthesized fatty acids and total cholesterol formed during ^2^H_2_O labeling period was assessed using a combinatorial model of polymerization biosynthesis, as described previously [18,45,46]. Mass isotopomer distribution analysis (MIDA) along with body ^2^H_2_O enrichment, representing the precursor pool enrichment (p), is used to determine the theoretical maximum enrichment of each analyte [18,42,43,45]. Using the measured deuterium enrichments, fractional and absolute contributions from DNL are then calculated. The value for f DNL represents the fraction of total triglyceride or phospholipid palmitate in the depot derived from DNL during the labeling period, and absolute DNL represents grams of palmitate synthesized by the DNL pathway.

### 4.13. RNAseq

RNA was isolated using standard Trizol protocol, and RNA concentrations were obtained using a Nanodrop. Library preparation was performed using Kapa Biosystems mRNA Hyper Prep Kit. Sequencing was performed using NovaSeq, mode SR100 through the Vincent J. Coates Genomic Sequencing Core at University of California, Berkeley. Trimmed fastq reads were then aligned to the mouse genome and analyzed using Qiagen CLC Workbench Software. Differentially expressed genes were initially separated based on their direction (up/down). We then looked at which processes were enriched given the differentially expressed gene set with Gorilla [41]. We did a negative-log transform of the *p*-values for each significant enrichment and generated a figure using the Matplotlib package in Python.

### 4.14. Electron Microscopy

Electron microscopy was performed at the UC Berkeley Electron Microscope Laboratory. Samples were no larger than 0.5 mm and were agitated at each step. Samples were fixed for 1 h in 2% glutaraldehyde in 0.1 M sodium cacodylate buffer, pH 7.2, rinsed for 10 min in 0.1 M sodium cacodylate buffer, pH 7.2, three times, put in 1% osmium tetroxide in 1.6% potassium ferricyanide for 1 h, and rinsed for 10 min in 0.1 M sodium cacodylate buffer, pH 7.2, three times. Then, they were dehydrated in acetone: 35% acetone 10 min, 50% acetone 10 min, 70% acetone 10 min, 80% acetone 10 min, 95% acetone 10 min, 100% acetone 10 min, 100% acetone 10 min, 100% Acetone 10 min. Samples were then infiltrated with 2:1 acetone:resin (accelerator) for 1 h, 1:1 acetone:resin for 1 h, 75% acetone 25% resin, overnight. The next morning, samples were put in pure resin for 1 h, changed three times, and then in pure resin plus accelerator for 1 h. Samples were embedded into molds at 60 °C for 2 days with pure resin and accelerator. Samples were then visualized via the TECNAI 12 TEM.

### 4.15. Hematoxylin and Eosin Staining

Hematoxylin and eosin staining was performed at the UCSF Biorepository and Tissue. Images were collected with a Zeiss Plan-Apochromat 20×/0.8 NA (WD = 0.55 mm) M27 Biomarker Technology Core. Imaging was conducted in a Zeiss Axio Scan.Z1 whole slide scanner objective lens in the brightfield mode with Hitachi HV-F202 camera.

## Figures and Tables

**Figure 1 ijms-23-01073-f001:**
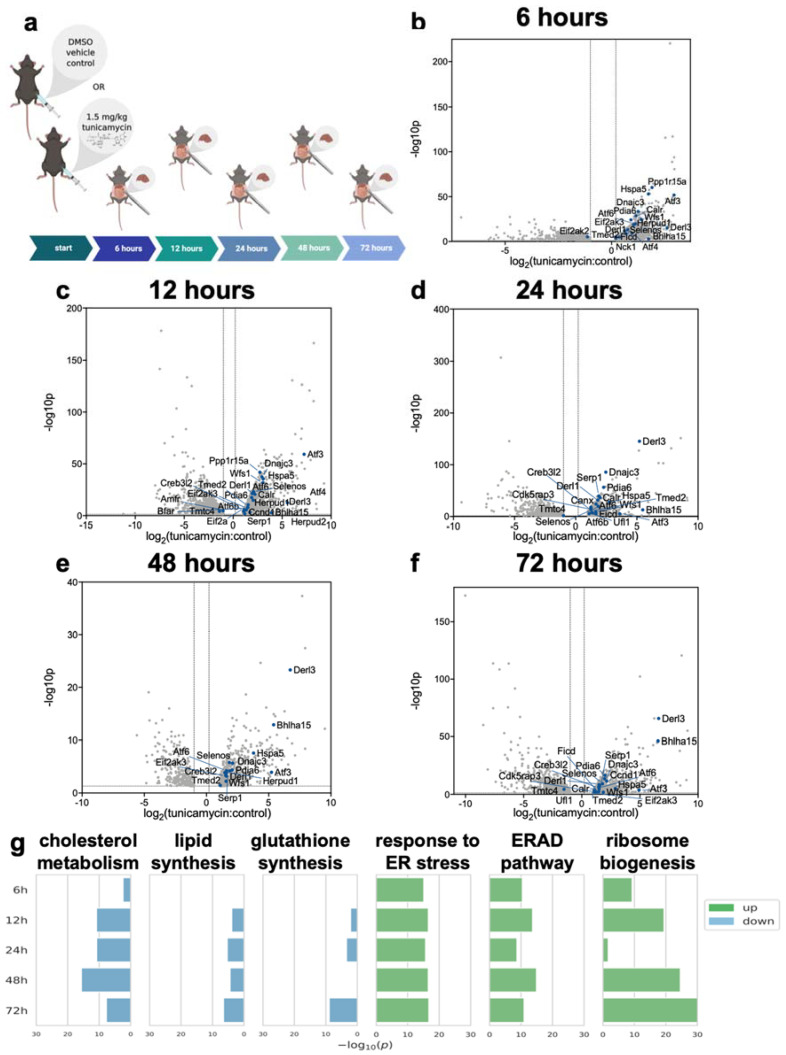
(**a**) Experimental overview. Mice (*n* = 5 per group) were treated with either DMSO or 1.5 mg/kg tunicamycin. Tissues were taken 6–72 h post-treatment. (**b**–**f**) Volcano plot of all genes for which RNA-seq measured expression. Points expressed as log2 fold-change tunicamycin treated/control on x-axis and -log10 (*p*-value), obtained from 2-tail *t*-test, on *y*-axis. (**g**) GO (gene ontology) analysis for genes for which tunicamycin treatment significantly changed gene expression. GO-terms indicate groups for which a significant number of genes where changed in relation to the remainder of the data set for each time point. GO with significant decreased gene expression: cholesterol metabolic process (GO: 0008203), lipid biosynthetic process (GO: 0008610), glutathione biosynthetic process (GO: 0006749). GO with significant increased gene expression: response to ER stress (GO: 0034976), ERAD pathway (GO: 0036503), ribosome biogenesis (GO: 0042254). All gene abbreviations used are consistent with Uniprot.

**Figure 2 ijms-23-01073-f002:**
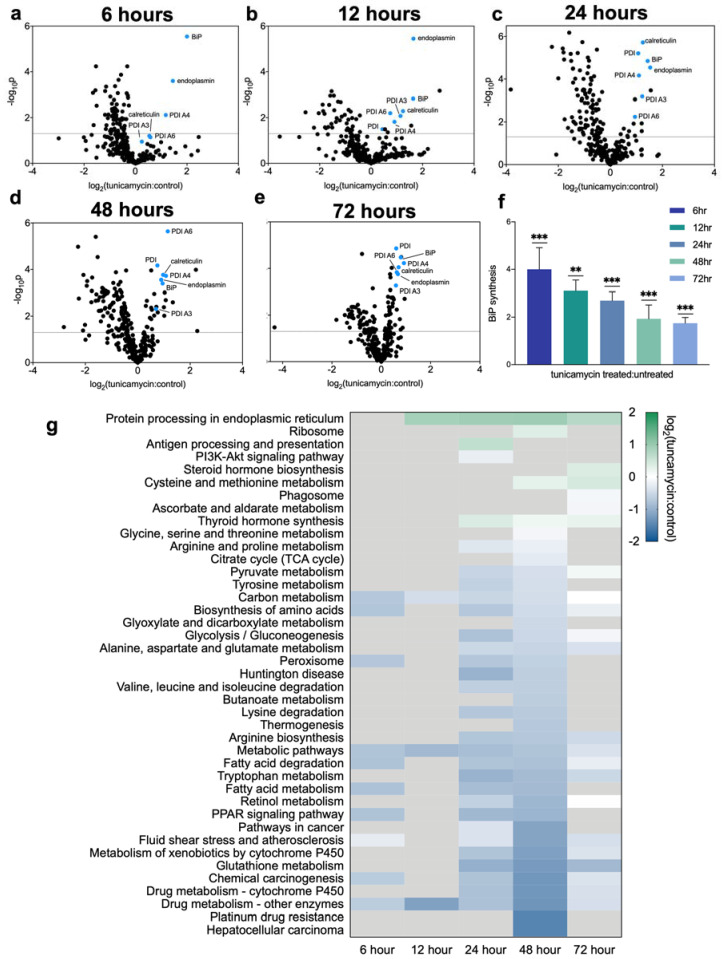
(**a**–**f**) Volcano plot of all proteins for which fractional synthesis rates were measured 12–72 h post tunicamycin treatment. Points expressed as log2 fold-change tunicamycin treated/control on *x*-axis and −log10 (*p*-value), obtained from 2-tail *t*-test, on *y*-axis. (**f**) Ratio of protein translation rates of BiP in tunicamycin treated/control. Ns = no significance, ** = <0.01, *** = <0.001. (**g**) KEGG-pathway analysis for fractional synthesis rates of significant (*p* = <0.05 per 2-tail *t*-test) proteins from tunicamycin treated/control. *n* = at least 5 proteins per pathway. All protein abbreviations used are consistent with Uniprot; accession numbers can be found in the Appendix A.

**Figure 3 ijms-23-01073-f003:**
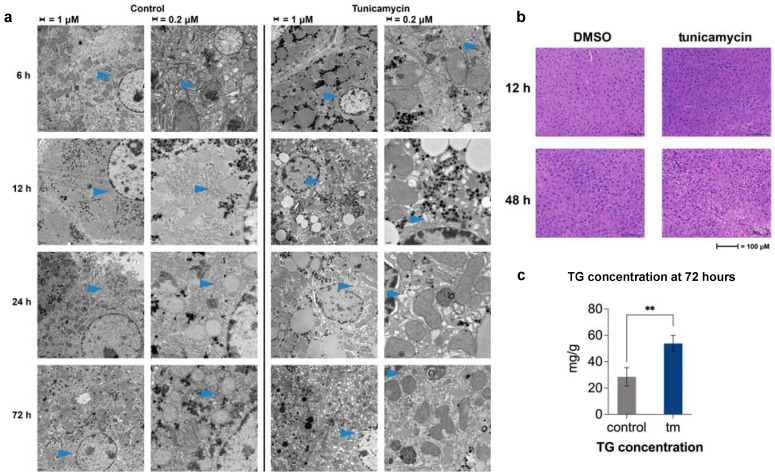
(**a**) Electron microscopy (TEM) images of liver sections 12–72 h post DMSO (control) or with 1.5 mg/kg tunicamycin treatment. Arrows point out ER in both treated and control to highlight changes in morphology. (**b**) H&E staining of liver sections 12 and 48 h post DMSO or with 1.5 mg/kg tunicamycin treatment. (**c**) Concentration of triglycerides in mouse liver lysate post DMSO (control) or with 1.5 mg/kg tunicamycin treatment. (Tm = tunicamycin-treated). ** = <0.01.

**Figure 4 ijms-23-01073-f004:**
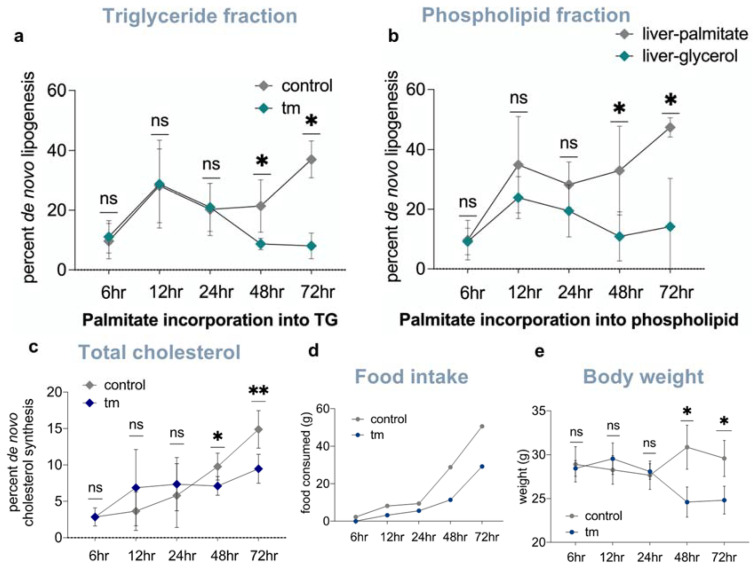
(**a**) De novo lipogenesis rates of palmitate incorporated into triglycerides in control and tunicamycin treated mice 12–72 h post treatment (*n* = 5 per group). (**b**) De novo lipogenesis rates of palmitate incorporated into phospholipids in control and tunicamycin treated mice 12–72 h post treatment (*n* = 5 per group). (**c**) De novo cholesterol synthesis (free and esterified) rates in control and tunicamycin treated mice 12–72 h post treatment (*n* = 5 per group). (**d**) Food intake in control and tunicamycin treated mice 12–72 h post treatment (*n* = 5 per group). (**e**) Average body weight of control and tunicamycin treated mice 12–72 h post treatment (*n* = 5 per group). ns = no significance, * = <0.05, ** = <0.01.

**Figure 5 ijms-23-01073-f005:**
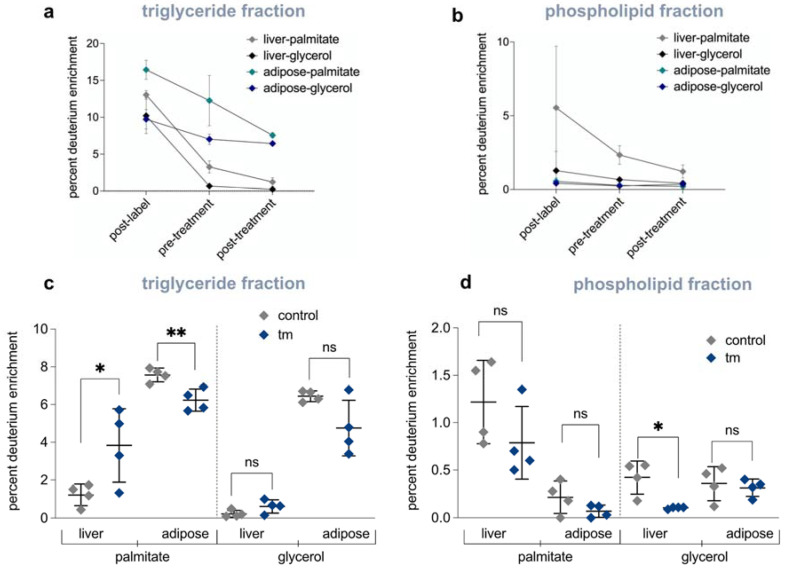
(**a**) Pre-labeling of triglycerides in control (DMSO) mice: percent deuterium enrichment of palmitate incorporated into triglycerides after 7 weeks of deuterium labeling (post-label), after a 2 week label free period (pre-treatment), and post 72 h treatment period (*n* = 3 per group). (**b**) Pre-labeling of phospholipids in control (DMSO) mice: percent deuterium enrichment of palmitate incorporated into phospholipids after 7 weeks of deuterium labeling (post-label), after a 2 week label free period (pre-treatment), and post 72 h treatment period (*n* = 3 per group). (**c**) Percent deuterium incorporated into palmitate and glycerol of triglycerides found in the liver or adipose after control (DMSO) or 1.5 mg/kg tunicamycin treatment after 72 h (*n* = 4 per group). (**d**) Percent deuterium incorporated into palmitate and glycerol of phospholipids found in the liver or adipose after control (DMSO) or 1.5 mg/kg tunicamycin treatment after 72 h (*n* = 4 per group). ns = no significance, * = <0.05, ** = <0.01.

**Figure 6 ijms-23-01073-f006:**
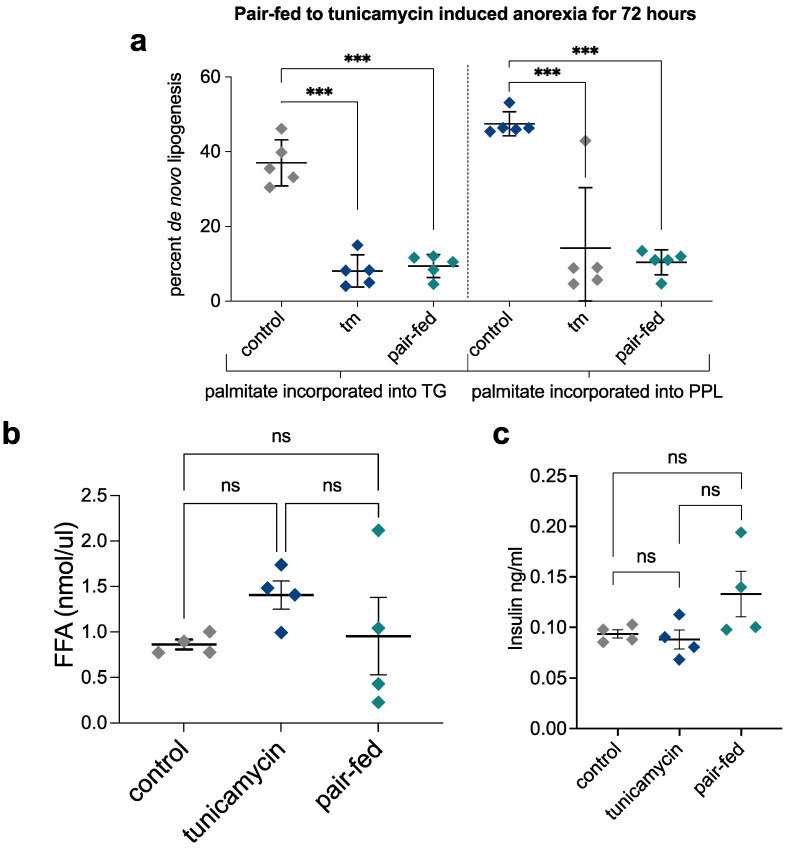
Hepatic de novo lipogenesis, free-fatty acid (FFA), and insulin at 72 h post treatment. (**a**) Hepatic de novo lipogenesis in pair-fed mice matched to tunicamycin treated mice. (**b**) Serum FFA at 72-h post treatment in control, tunicamycin, and pair-fed mice. (**c**) Serum insulin at 72 h post treatment in control, tunicamycin, and pair-fed mice. *N* = 5 mice per group. ns = no significance, *** = <0.001.

## Data Availability

The data that support the findings of this study are openly available as Appendix A to our publication.

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
