# Peer review of "ER Unfolded Protein Response in Liver In Vivo Is Characterized by Reduced, Not Increased, De Novo Lipogenesis and Cholesterol Synthesis Rates with Uptake of Fatty Acids from Adipose Tissue: Integrated Gene Expression, Translation Rates and Metabolic Fluxes"

_ijms, 2022, doi:10.3390/ijms23031073_

Round 1
Reviewer 1 Report
The authors did well study on an entrancing topic. I think this study will have a high citation rate. I recommend that the article be accepted.
Thank you for giving me the opportunity to review the article again. I carefully reviewed the article again. The study was not supported by biochemical parameters such as AST, ALT, GGT, and glutamate dehydrogenase related to hepatic steatosis. However, the authors conducted a study to shed light on hepatic lipid accumulation and presented new information not previously reported in the literature. That's why I believe the study offers valuable findings. I have some minor corrections suggestions. 1. In the introduction section, the mechanisms of hepatic steatosis should be briefly mentioned. 2. The last part of the introduction should be the first paragraph of the discussion (Paragraph on Line 70 in the introduction [We report here a striking decline in....]). 3. In the last part of the introduction, the purpose of the study should be clearly stated. 4. In the first part of the method, the total number of animals and how many animals were selected for each group should be stated. 5. The statistics part should be written in detail. The tests used for statistics should be mentioned. 6. The word should be written clearly before using the abbreviation. 7. There are some grammatical errors in the article, and some sentences are difficult to understand. It is recommended to review the language of the article.
Author Response
-
Thank you to reviewer 1 for thoughtful analysis of our paper and suggested improvements. We believe our paper is stronger with the suggested incorporated changes, for which we are grateful.As requested:
1) We elaborated on hepatic steatosis and UPRER involvement in the background.
2) We moved the last paragraph of the background to the first part of the discussion.
3) We added the purpose of the study at the end of the background.
4) We added numerical values for mice in the methods.
5) We added details about statistical analysis in the methods section.
6) We added descriptions for abbreviations used (including ERAD and KEGG).
7) We reviewed the article for grammatical errors and sentence structure.
Reviewer 2 Report
This work offers a fundamental step in understanding the metabolic responses to the initiation of the unfolded protein response in the endoplasmic reticulumin in vivo, induced by tunicamycin administration (in mouse). The Authors have accurately measured synthesis rates of proteins, fatty acids, and cholesterol at different time points. Contemporarily, they have performed RNA-seq on the liver tissue to compare mRNA levels to rates of protein synthesis and lipogenic pathway fluxes.
Results demonstrate that, hepatic de novo lipogenesis and cholesterogenesis are markedly reduced, as well as the mRNA levels and synthesis rates of lipogenic proteins. In this context, lipid accumulation in the liver does not derive from hepatic lipogenesis but from adipose tissue, as observed by greater mobilization of labeled lipids. Finally, the redistribution of fatty acids to liver does not correlate to altered plasma insulin levels, adipose lipolysis or food intake.
I do have a few minor concerns and revisions, which are listed below. Once these are addressed, I strongly recommend acceptance of this manuscript for publication.
- Since this study has been done using only tunicamycin among possible UPR(ER)-inducing compounds, and these compounds have different modes of action, the Authors should discuss in detail the choice of tunicamycin in the text, and comment how the use of other UPR(ER)-inducing compounds might affect their findings.
- Mice were injected IP using tunicamycin dissolved in DMSO or DMSO control at a certain percentage. The authors should specify adequately the percentage of DMSO in the text during the different analyses.
- Supplementary Table 1 reports all the Individual protein synthesis rate ratios of tunicamycin/control mice, but not the specific acronym of each protein and not the process in which the target is involved (e.g., cholesterol metabolism and other processes reported in Figure 2g). Figure 1 (b-f) and Figure 2 (a-f) report only the specific target acronym but not their extensive name in the legend. Finally, some acronyms are not defined explicitly in the text (e.g., line 42-43, line 90, line 110, line 113).
To help the readers, the Authors should revise the Supplementary Table 1, the legend of Figure 1 (b-f) and Figure 2 (a-f), and the text, accordingly.
- The Authors did not observe significant differences between control, tunicamycin-treated and pair-fed (to tunicamycin) groups for plasma insulin at 72 hours post treatment (Figure 6c). The Authors should comment in the text the observed values (are they in line with values in the fasted state or during digestion?) and why they have not reported values of insulin at different time points. This would help the readers to follow the discussion.
Author Response
Thank you to reviewer 2 for thoughtful analysis of our paper and suggested improvements. We believe our paper is stronger with the suggested incorporated changes, for which we are grateful.
As requested:
- We added discussion of our choice of tunicamycin in the first paragraph of the results section.
- We elaborated on amount of DMSO injected per mouse in the methods section.
- Figure 1 reports genes, not proteins, and does not correlate with supplementary table 1. Genes acronyms used were in alignment with Uniprot. Figure 2 reports protein names which correlate to table 2. Acronyms used are also in alignment with Uniprot, and accession numbers and processes which targets are involved in are available in the supplementary excel file attached to our paper. We updated the acronyms used on lines 42-43 (IRE1a, BiP, PERK, ATF6), 90 (ERAD), and 113 (KEGG). We updated the legends for figures 1 and 2 to provide clarity for both gene and protein abbreviations.
- We added discussion to the results (lines 237-239) to discuss measurement of plasma insulin.